# Implementation of EHMRG Risk Model in an Italian Population of Elderly Patients with Acute Heart Failure

**DOI:** 10.3390/jcm11112982

**Published:** 2022-05-25

**Authors:** Lorenzo Falsetti, Vincenzo Zaccone, Emanuele Guerrieri, Giulio Perrotta, Ilaria Diblasi, Luca Giuliani, Linda Elena Gialluca Palma, Giovanna Viticchi, Agnese Fioranelli, Gianluca Moroncini, Adolfo Pansoni, Marinella Luccarini, Marianna Martino, Caterina Scalpelli, Maurizio Burattini, Nicola Tarquinio

**Affiliations:** 1Internal and Subintensive Medicine Department, Azienda Ospedaliero-Universitaria “Ospedali Riuniti” di Ancona, 60100 Ancona, Italy; 2Emergency Medicine Residency Program, Marche Polytechnic University, 60100 Ancona, Italy; e.guerrieri93@gmail.com (E.G.); ilariadiblasi@gmail.com (I.D.); luca.giuliani33@gmail.com (L.G.); 3General and Emergency Surgical Clinic, Azienda Ospedaliero-Universitaria “Ospedali Riuniti”, 60126 Ancona, Italy; info@giulioperrotta.com; 4Internal Medicine Residency Program, Marche Polytechnic University, 60126 Ancona, Italy; elena.giallucapalma@gmail.com; 5Neurologic Clinic, Marche Polytechnic University, 60126 Ancona, Italy; viticchi.g@gmail.com; 6Internal Medicine Department, INRCA-IRCCS di Osimo, 60027 Ancona, Italy; agnese.fioranelli@gmail.com (A.F.); mariannamartino88@gmail.com (M.M.); caterina.scalpelli@gmail.com (C.S.); m.burattini@inrca.it (M.B.); n.tarquinio@inrca.it (N.T.); 7Clinica Medica, Marche Polytechnic University, 60124 Ancona, Italy; g.moroncini@univpm.it; 8Emergency Department, INRCA-IRCCS di Osimo, 60027 Ancona, Italy; a.pansoni@inrca.it (A.P.); m.luccarini@inrca.it (M.L.)

**Keywords:** EHMRG, acute heart failure, prognosis, emergency department

## Abstract

Acute heart failure (AHF) is a cardiac emergency with an increasing incidence, especially among elderly patients. The Emergency Heart failure Mortality Risk Grade (EHMRG) has been validated to assess the 7-days AHF mortality risk, suggesting the management of patients admitted to an emergency department (ED). EHMRG has never been implemented in Italian ED nor among elderly patients. We aimed to assess EHMRG score accuracy in predicting in-hospital death in a retrospective cohort of elderly subjects admitted for AHF from the ED to an Internal Medicine Department. We enrolled, in a 24-months timeframe, all the patients admitted to an Internal Medicine Department from ED for AHF. We calculated the EHMRG score, subdividing patients into six categories, and assessing in-hospital mortality and length of stay. We evaluated EHMRG accuracy with ROC curve analysis and survival with Kaplan–Meier and Cox models. We collected 439 subjects, with 45 in-hospital deaths (10.3%), observing a significant increase of in-hospital death along with EHMRG class, from 0% (class 1) to 7.7% (class 5b; *p* < 0.0001). EHMRG was fairly accurate in the whole cohort (AUC: 0.75; 95%CI: 0.68–0.83; *p* < 0.0001), with the best cutoff observed at >103 (Se: 71.1%; Sp: 72.8%; LR+: 2.62; LR-: 0.40; PPV: 23.0%; NPV: 95.7%), but performed better considering the events in the first seven days of admission (AUC: 0.83; 95%; CI: 0.75–0.91; *p* < 0.0001). In light of our observations, EHMRG can be useful also for the Italian emergency system to predict the risk of short-term mortality for AHF among elderly patients. EHMRG performance was better in the first seven days but remained acceptable when considering the whole period of hospitalization.

## 1. Introduction

Acute heart failure (AHF) represents a cardiac emergency that is often managed in the emergency departments (ED) and then treated, according to its severity and prognosis, in ED short stay areas, internal medicine or cardiology wards, or—in the most severe cases—in subintensive or intensive care units [1]. Both de novo AHF and acutely decompensated heart failure (ADHF) are characterized by high mortality and increased hospitalization and re-hospitalization rates, which result in increased healthcare costs and significant mortality and morbidity [2,3].

Therefore, physicians involved in the decision-making process of AHF/ADHF are required to decide whether to admit, place them in short-stay or observation or directly discharge patients with AHF/ADHF. A relevant proportion of these subjects are directly discharged from ED, and this subpopulation is at the highest risk of short-term adverse events, including early re-hospitalization and death [4,5]. Physician’s clinical gestalt is often adopted to assess patients′ prognosis and decide the disposition, although it is widely recognized as a non-accurate method [6,7] that can lead to unnecessary hospitalizations or, on the other side, to serious adverse events related to an early discharge. Of note, there is a substantial overlap in the prognostic profiles of ED patients who are subsequently discharged or hospitalized. This phenomenon can be explained by the fact that diagnostic approaches are often not clear and linear, with the risk of admitting low-risk patients and discharging subjects with only an apparent low risk, thus increasing the risk of serious events. Therefore, several prognostic algorithms have been studied to improve AHF prognostic evaluation in the ED [8]. 

Several factors have been considered in AHF prognosis, and several authors suggested evaluating six fundamental dimensions in these patients: blood pressure, heart rate, heart rhythm, precipitating factors, comorbid conditions, and clinical severity [9]. Previous prognostic studies have focused on AHF/ADHF patients who have been hospitalized but have resoundingly excluded those discharged from the ED [10,11,12,13], although it should be pointed out that those discharged from the ED may be a substantial proportion of all patients with HF and may also be at significant risk of acute mortality [7]. This observation limits the use of previous hospitalization-based risk algorithms in the broader context of ED. Thus, there is no guarantee that AHF risk algorithms developed in patients already admitted will identify those who can be safely discharged from ED [14].

Most of the risk stratification tools for AHF were designed in cohorts of cardiology inpatients [10,12], therefore, not being applicable to other populations, like ED patients [15,16]. Among the others, the most adopted tools to stratify AHF prognosis in clinical practice are the OPTIMIZE-HF and the ADHERE risk scales, which were specifically designed to improve in-hospital management of patients with AHF/ADHF [17,18]. However, since the derivation and validation cohorts were used in specific clinical settings, the possibilities to extend these clinical prediction rulers (CPRs) in other clinical settings, such as in ED or Internal Medicine, and in the short-term prognosis are limited [19]. Recently, several authors proposed specific risk scores for the ED to fill this gap: among the others, the Ottawa Heart Failure Risk Scale (OHFRS), the MEESSI risk score, and the Emergency Heart Failure Mortality Risk Grade (EHMRG) are the most studied and validated scores in this setting [20,21,22]. EHMRG is based on commonly adopted parameters but is very complex to calculate without a smartphone app and can only predict mortality, while the other CPRs are usually simpler to calculate but adopt some parameters that are more difficult to collect during the primary assessment of a critical patient [20,22].

EHMRG was originally engineered and validated in Canada, then validated in other countries, such as Spain, showing similar performances [23]. Italian patients are often older and less burdened by atherosclerosis than American populations, while they are more clinically and socially similar to Spanish subjects. However, a study on the implementation of this score in the Italian sanitary system and in the Italian geriatric population is missing. With this paper, we aimed to assess the efficacy of EHMRG score in predicting in-hospital death in a cohort of elderly Italian subjects admitted for AHF in the ED.

## 2. Materials and Methods

Background: Methods of this study have already been described elsewhere [24]. The INRCA-IRCSS (National Institute for Care and Research in Aging, Ancona, Italy) Hospital of Osimo (Ancona, Italy) is a primary hospital specializing in the acute care of elderly subjects. Regarding AHF/ADHF, patients admitted to the ED can directly access the Internal Medicine department. However, AHF associated with other acute conditions (STE/NSTE myocardial infarction requiring revascularization, brady- or tachyarrhythmias requiring pacing or other device therapy, acute valvular and cardiac diseases requiring cardiac surgery, or severe associated critical illnesses, such as pulmonary embolism, septic or cardiogenic shock that require admission to intensive care unit) are directly sent or to the cardiac care unit of the same institute or to a tertiary-care hospital after pre-hospital or ED evaluation. As such, very-high risk patients are not present in this sample, which mainly includes subjects affected by ADHF or de novo AHF due to medical conditions, such as arrhythmias, hypertensive crises, and other causes. ADHF/AHF was diagnosed by the attending physician according to the ESC 2016 guidelines that were current at the moment of the study [25].

Ethical Issues: This study was authorized on 6 May 2021 by the INRCA-IRCSS Ethical Committee (CE INRCA, protocol n° 21011/21-CE) and then approved by INRCA Hospital (protocol n° 193, 26 May 2021). All patients gave their informed consent and were treated according to the guidelines current at the time of the study. We followed the Declaration of Helsinki Ethical Principles for Medical Research Involving Human Subjects. 

Enrolment, Inclusion, and Exclusion Criteria: In a 24-months timeframe (1 January 2018–30 December 2019), we retrospectively enrolled all the patients aged 60 or more years and assessed in the ED, and then admitted to the Internal Medicine Department with ADHF/AHF diagnosis. We adopted the same exclusion criteria of the original EHMRG study [22]: (i) transfer from another department (ICU, Cardiology, Pneumology) or direct admission from the heart failure ward, (ii) patients in end-of-life care due to active cancer or other terminal comorbidities, (iii) dialysis-dependent subjects. We also excluded patients with incomplete data that did not allow us to correctly calculate the EHMRG score. 

Data Collection: We gathered history and vital signs at the ED arrival. For each patient, we collected: age, modality of ED transport, systolic blood pressure (SBP), heart rate (HR), oxygen saturation (SpO2), serum creatinine, serum potassium, serum troponin, presence of active cancer, and metolazone use at home. From these items, we calculated the absolute EHMRG score according to its original definition, as shown in Table 1, and then recategorized the subjects into the six EHMRG categories (Class 1: −49.1; Class 2: from −49.0 to −15.9; Class 3: from −15.8 to 17.9; Class 4: from 18.0 to 56.5; Class 5a: from 56.6 to 89.3; Class 5b: 89.4). Last, we evaluated the length of admission and in-hospital mortality. 

Statistical Analysis: We presented continuous variables with normal distribution as mean and standard deviation (SD) and compared them with a t-test for independent variables. We synthesized non-normally distributed variables with median and interquartile range [IQR], adopting the Mann–Whitney U test. We presented categorical variables as absolute number and percent, comparing them with the chi-squared test. We evaluated the EHMRG accuracy for in-hospital death with ROC curve analysis, considering both the events (discharge or in-hospital mortality) during the whole time of observation and the events observed in the first seven days. We identified the best cutoff point with a critical ROC curve assessment and adopted the Youden Index. We performed a univariate test to select covariates, choosing the ones associated with the outcome of interest at a level of *p* < 0.10 and excluding the items already considered in the EHMRG score to avoid multicollinearity. Last, we performed a Cox multivariate model considering days of admission as the time variable, in-hospital death as the event variable, EHMRG category as the main predictor, and the covariates selected by univariate test, both in the full sample and in the seven days sample. We considered as significant all the differences at a level of *p* < 0.05. We performed the analysis with SPSS 13.0 for Windows Systems (SPSS Inc., Chicago, IL, USA).

## 3. Results

We obtained a cohort of 439 subjects with 45 (10.3%) deaths. Baseline characteristics of the full cohort and of the events observed at seven days are synthesized in Table 2, while differences between surviving and non-surviving subjects are shown in the Appendix A.

EHMRG predicted with fair accuracy in-hospital death in the whole cohort when treated as continuous (AUC: 0.754; 95%CI: 0.68–0.83; *p* < 0.0001) and categorial (AUC: 0.727; 95%CI: 0.66–0.80; *p* < 0.0001), as shown in Figure 1A.

Analyzing the ROC curve drawn from the continuous EHMRG variable, we observed, at the optimal cutoff of 103, a sensitivity of 71.1% (95%CI: 55.7–83.6%), a specificity of 72.8% (95%CI: 68.2–77.2%), a positive likelihood ratio of 2.62 (95%CI: 2.0–3.4), a negative likelihood ratio of 0.40 (95%CI: 0.2–0.6), a positive predictive value of 23.0% (95%CI: 16.3–30.9%), a negative predictive value of 95.7% (92.7–97.7%). On the other hand, analyzing the ROC curve drawn from the categorical EHMRG variable, we observed, at the optimal cutoff of 3, a sensitivity of 95.6% (95%CI: 84.9–99.5%), a specificity of 22.1% (95%CI: 18.1–26.5%), a positive likelihood ratio of 1.23 (95%CI: 1.1–1.3), a negative likelihood ratio of 0.20 (95%CI: 0.05–0.8), a positive predictive value of 12.3% (95%CI: 9.0–16.2%), a negative predictive value of 97.8% (95%CI: 92.1–99.7%), which was comparable to the one observed in the original cohort. Prevalence of in-hospital death increased significantly across EHMRG categories, ranging from 0% in the first to 7.7% in the last category, as shown in Table 3 and in the Appendix A.

When reducing the observation period to the events of the first seven days (138 patients, 22 in-hospital deaths), EHMRG accuracy significantly increased for both continuous (AUC: 0.83; 95%CI: 0.75–0.91; *p* < 0.0001) and categorial (AUC: 0.80; 95%CI: 0.72–0.89; *p* < 0.0001) variables, as shown in Figure 1B. According to these results, the optimal cutoff was 4, holding a sensitivity of 90.9% (70.8–98.9%), a specificity of 55.2% (45.7–64.4%), a positive likelihood ratio of 2.03 (95%CI: 1.6–2.6), a negative likelihood ratio of 0.16 (95%CI: 0.04–0.6), a positive predictive value of 27.8% (95%CI: 17.8–39.7%) and a negative predictive value of 97.0% (95%CI: 89.5–99.6%), which was comparable to the one observed in the original cohort. Prevalence of in-hospital death increased significantly across EHMRG categories, ranging from 0% in the first to 12.3% in the last category: in this subgroup, the distribution of deaths was even more shifted towards higher EHMRG categories, with a significant difference in the distribution (*p* < 0.0001), as synthesized in Table 3 and in the Appendix A.

We observed in the univariate analysis that EHMRG class, NYHA category, BNP at the admission, and sex were associated with in-hospital death, thus, we maintained these variables in the final multivariate model. We did not include the other collected variables from the multivariate model since they were already considered in the EHMRG score: adding these features could have increased the risk of multicollinearity and the overinflation of the model. Cox regression analysis underlined that both in the whole cohort and in the seven-day events cohort, a one-unit increase in EHMRG category was associated with an increased hazard ratio (HR) of in-hospital death (HR: 2.85; 95%CI: 1.64–4.98; *p* < 0.0001). Furthermore, NYHA class was associated with an increased HR (HR: 2.84; 95%CI: 1.78–4.54; *p* < 0.0001), while BNP at the admission and sex became non-significant in the multivariate analysis. The full model is shown in the Appendix A. Considering the events in the first seven days, we observed similar results, as shown in the Appendix A.

## 4. Discussion

The EHMRG score has among its main strengths, the ease of use and the limited number of items required for the calculation that makes this CPR ideal for use in an Emergency Department. Particularly, EHMRG can be calculated only by vital parameters and data retrieved from the first contact between the patient and the triage nurse, and laboratory exams that are measured virtually in all the ED patients. However, despite its usability, its accuracy should be assessed, especially when translated to populations or emergency systems that differ from the original cohort, like all the clinical scores [24,26,27].

Our population varied from the original one [22] for a significantly older age (75.4 ± 11.4 years in the original cohort versus 84.6 ±7.7 years in our cohort), a different pattern of comorbidities, and a different sanitary system. Despite these differences, EHMRG maintained a similar accuracy in predicting in-hospital death in the first seven days (original cohort AUC: 0.81; 95%CI: 0.77–0.85; this cohort AUC: 0.83; 95%CI: 0.75–0.91). Our sample had characteristics similar to a recently published Spanish cohort [23], however, the accuracy was probably different for the different populations considered (ED patients in the Spanish cohort, ED subjects admitted to Internal Medicine in this sample), and the inclusion in that cohort of palliative patients.

When we extended the observation to the whole observation period, which was longer than the original one, we observed that EHMRG maintained a fair accuracy (AUC: 0.75; 95%CI: 0.68–0.83), suggesting the potential use of this score both for the ED and the Internal Medicine specialist.

However, at its best cutoffs (>103 in the continuous variable and >3 in the categorical variable), EHMRG showed a remarkable capacity to identify subjects at low risk of short-term events. According to our data, EHMRG accuracy in predicting death peaked when considering the events observed in the first seven days, which is the original timeframe for which EHMRG was designed to predict mortality. In this short period, EHMRG accuracy improved, as underlined by a significant AUC increase, from 75 to 83%, which is similar to the accuracy observed in the original and in the validation cohorts [22,23]. However, the most important datum in all the ROC curve analyses performed is the negative likelihood ratio, which was 0.20 when considering the whole cohort and 0.16 in the first seven days, underlining an important capacity of this score in excluding the events, best if in the very short-term. Moreover, the negative predictive values observed in this cohort are similar to the ones observed in the original derivation and validation cohort. The capacity of EHMRG in identifying very low-risk subjects seems to be especially useful in the ED, where the physician needs to accurately identify patients that will not undergo complications after an early discharge. On the other side, the capability of identifying low-risk subjects even after the strict seven days can be useful both for the ED physician to choose the best care setting, thus optimizing economic resources, and for the Internal Medicine specialist, who could be able to assess the short-term prognosis even during the admission to choose the most appropriate follow-up.

This study has its strengths: the population under exam comprises most of the patients admitted in our ED since the internal protocols, in the presence of a very small, short-stay area, did not allow to directly discharge the AHF patients. All the patients were followed up during the admission to Internal Medicine, and this is another important point since there were no patients lost after the ED disposition.

Limitations: the current study′s main limitation is related to its retrospective nature and the relatively small sample size. Multicentric, prospective studies with larger samples are necessary to obtain more reliable results in the same population. Moreover, to further reduce the risk of bias, it would be important to perform these studies by enrolling the subjects directly in the ED and following them up independently of their destination (discharge, short-stay, regular or subintensive ward), which could be, however, very difficult for the actual organization of the regional sanitary system, which allocates patients with different degrees of severity in different hospitals and departments. Another point is related to the lack of follow-up after discharge, which could be useful for assessing the early readmission rates. This point could represent a potential implementation for future studies assessing not only in-hospital mortality but also early readmissions rates: in fact, this composite outcome could be a more reliable marker of therapeutic failure in these patients. Last, this score does not suggest different therapeutic management during the in-hospital stay: a potential implementation for future studies could be to assess whether a different treatment according to this stratification could translate into better clinical outcomes.

## 5. Conclusions

The implementation of the EHMRG score can be useful to assess short-term prognosis in elderly patients with AHF evaluated in the ED and then managed in Internal Medicine in the Italian geriatric population. This score, however, seems to be more important to rule-out short-term mortality than to rule-in events, especially in lower-risk classes.

## Figures and Tables

**Figure 1 jcm-11-02982-f001:**
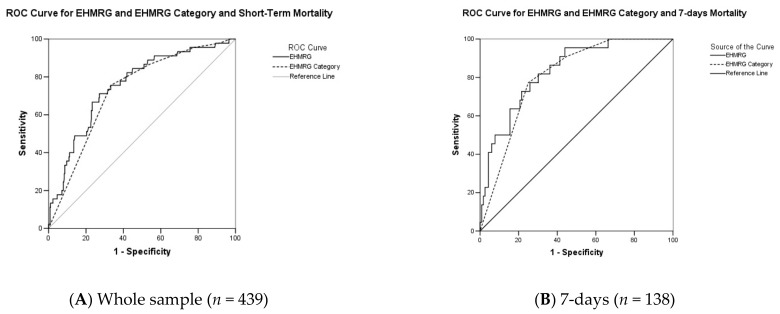
ROC curve analysis of continuous and categorial EHMRG for in-hospital death (panel (**A**): whole sample; panel (**B**): first 7 days of admission).

**Table 1 jcm-11-02982-t001:** EHMRG Score.

Variable	Units	Factor
Age	Years	2 × age
ED arrival by ambulance	If “yes”	+60
SBP	mmHg	−1 × SBP
Heart rate	beats/min	1 × HR
Oxygen saturation	%	−2 × Oxygen Saturation
Creatinine	mg/dL	20 × Creatinine
Serum potassium	4.0–4.5 mmol/L ≥4.6 mmol/L ≤3.9 mmol/L	0 +30 +5
Serum troponin	>ULN	+60
Active cancer	If “yes”	+45
Metolazone at home	If “yes”	+60
Adjustment factor		+12
Total		

Legend: EHMRG = Emergency Heart Failure Mortality Risk Grade; ED = emergency department; SBP = systolic blood pressure; ULN = upper limit of normal.

**Table 2 jcm-11-02982-t002:** Baseline characteristics of the sample.

Clinical Variables	Full Cohort (*n* = 439)	7 Days (*n* = 138)
Age, years, (±SD)	84.6 (±7.7)	84.1 (± 8.3)
Males (*n*, %)	180 (41.0%)	64 (46.4%)
In-hospital death (*n*, %)	45 (10.3%)	22 (15.9%)
NYHA class, [IQR]	4 [1]	3 [1]
Length of hospitalization, days, [IQR]	10 [7]	--
BNP on admission, pg/mL, [IQR]	600.5 [805]	560.5 [846]
SBP, mmHg, (±SD)	127.5 (±28.1)	128.0 (±28.2)
HR, bpm, (±SD)	89.4 (±24.6)	90.4 (±23.9)
SpO2, %, (±SD)	91.8 (±7.3)	92.0 (±7.07)
Creatinine, mg/dl, (±SD)	1.6 (±1.0)	1.45 (±0.99)
Potassium, mmol/l, (±SD)	4.00 (±0.69)	4.04 (±0.65)
Out of range Potassium, (*n*, %)	180 (41.1%)	74 (53.6%)
Troponin, ng/mL, [IQR]	0.05 [0.10]	0.05 [0.11]
Increased troponin, (*n*, %)	204 (46.5%)	63 (45.7%)
ED arrival by ambulance, (*n*, %)	284 (64.7%)	83 (60.1%)
Active cancer, (*n*, %)	77 (17.9%)	16 (11.6%)
Metolazone use, (*n*, %)	11 (2.6%)	1 (0.72%)
EHMRG, [IQR]	69 [98.4]	60,8 [99.3]
EHMRG Class, [IQR]	5 [2]	5 [3]
AHF characteristics		
ADHF (*n*, %)	370 (84.2%)	109 (78.9%)
AHF de novo (*n*, %) Arrhythmia Hypertensive crisis Other	36 (8.20%) 21 (4.78%) 12 (2.73%)	11 (7.97%) 12 (8.69%) 6 (4.34%)

Legend: AHF = acute heart failure; ADHF = acutely decompensated heart failure; BNP = brain-derived natriuretic peptide; EHMRG = Emergency Heart Failure Mortality Risk Grade; HR = heart rate; IQR = interquartile range; SBP = systolic blood pressure; NYHA = New York Heart Academy; SD = standard deviation; SpO2 = oxygen saturation.

**Table 3 jcm-11-02982-t003:** Distribution of in-hospital death according to EHMRG score (*p* < 0.0001).

EHMRG Category In-Hospital Death	Full Sample (*n* = 439)	7-Days Observation (*n* = 138)
EHMRG Category 1 (*n*, %)	0 (0.0%)	0 (0.0%)
EHMRG Category 2 (*n*, %)	1 (0.2%)	0 (0.0%)
EHMRG Category 3 (*n*, %)	1 (0.2%)	0 (0.0%)
EHMRG Category 4 (*n*, %)	4 (4.1%)	2 (1.4%)
EHMRG Category 5a (*n*, %)	5 (1.1%)	3 (2.2%)
EHMRG Category 5b (*n*, %)	34 (7.7%)	17 (12.3%)
Total	45 (10.3%)	22 (15.9%)

Legend: EHMRG = Emergency Heart Failure Mortality Risk Grade.

## Data Availability

The data presented in this study are available on request from the corresponding author. The data are not publicly available due to privacy issues.

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
