# Peer review of "Implementation of EHMRG Risk Model in an Italian Population of Elderly Patients with Acute Heart Failure"

_jcm, 2022, doi:10.3390/jcm11112982_

Round 1

Reviewer 1 Report

This study is very interesting and useful for the management of ED patients. This is a validation of EHMRG risk model in Italian population. The overall design and method of the study are quite good.

Introduction: In my opinion the abstract needs to focus more on the main conclusions and briefly present the method (in this abstract the method is very extended) in order to be more easily understood.

Introduction: The first two paragraphs are very general and not related substantially to subject. The rest is very good.

Material-methods: Nothing to comment, well-explained.

Results: The first sentence of the last paragraph (lines 214-217) is not well expressed. Please make this part more conceivable.

Discussion: Please compare your results with other researches in parameters such as specificity, negative likelihood ratio, sensitivity. Provide more clues about the Italian population in comparison to other populations.

Limitations: Well presented part

Conclusions: Good

I want to ask if you have recorded the re-admissions after early discharge?

If not , please comment on this.

Overall this manuscript is well-written but it is a validation and not a new method. The method of validation is very and the adjustment in Italian population very useful.

Author Response

Q1. This study is very interesting and useful for the management of ED patients. This is a validation of EHMRG risk model in Italian population. The overall design and method of the study are quite good.

A1. We thank the reviewer for this overall positive comment.

Q2. Introduction: In my opinion the abstract needs to focus more on the main conclusions and briefly present the method (in this abstract the method is very extended) in order to be more easily understood.

A2. In the revised version, we have improved the abstract according to the reviewer’s comments.

Q3. Introduction: The first two paragraphs are very general and not related substantially to subject. The rest is very good.

A3. We thank the reviewer for this suggestion and for this positive comment. In the revised version, we have reduced the first two paragraphs into a smaller one.

Q4. Material-methods: Nothing to comment, well-explained.

A4. We thank the reviewer for this overall appreciative comment.

Q5. Results: The first sentence of the last paragraph (lines 214-217) is not well expressed. Please make this part more conceivable.

A5. We thank the reviewer for this comment. In the revised version, we have rephrased the sentence in order to improve its readability.

Q6. Discussion: Please compare your results with other researches in parameters such as specificity, negative likelihood ratio, sensitivity. Provide more clues about the Italian population in comparison to other populations.

A3. We thank the reviewer for this suggestion. In the revised version, we have improved the discussion.

Q7. Limitations: Well presented part

A7. We thank the reviewer for this overall appreciative comment.

Q8. Conclusions: Good

A8. We thank the reviewer for this overall appreciative comment.

Q9. I want to ask if you have recorded the re-admissions after early discharge? If not , please comment on this.

A9. We thank the reviewer for this suggestion. However, we do not have this information. We have added a sentence discussing this point in the “study limitations” part, suggesting this interesting point for further implementations.

Reviewer 2 Report

Implementation of EHMRG risk model in an Italian population of elderly patients with acute heart failure

The aim of the study is to validate the Emergency Heart Failure Mortality Risk Grade system in patients admitted for acute heart failure in an emergency department. The main weakness of the study is that it is a retrospective study, with a limited number of patients included. The conclusions are appropriate. In my opinion there is no correlation with the clinic. Knowing how to predict the mortality rate is only meaningful if related to therapy. Did the identification of a predictive system allow a reduction in mortality thanks to an optimization of the anti-decopensation therapy?

Author Response

Q10. Overall this manuscript is well-written but it is a validation and not a new method. The method of validation is very and the adjustment in Italian population very useful.

A10. We thank the reviewer for this overall positive comment. We chose to validate an existing method instead of producing a new one because we deem that EHMRG risk model is very easy and a larger validation could be useful for the practicing physician.

Q11. The aim of the study is to validate the Emergency Heart Failure Mortality Risk Grade system in patients admitted for acute heart failure in an emergency department. The main weakness of the study is that it is a retrospective study, with a limited number of patients included.

A11. We agree with the reviewer: this is the main limitation of this study. We have underlined this point in the study limitations section.

Q12. The conclusions are appropriate. In my opinion there is no correlation with the clinic. Knowing how to predict the mortality rate is only meaningful if related to therapy. Did the identification of a predictive system allow a reduction in mortality thanks to an optimization of the anti-decopensation therapy?

A12. We thank the reviewer for this interesting point. This is not the objective of this study, since EHMRG aims at predicting early mortality at the patient’s arrival in the ED, thus suggesting an optimization of the management of the patient (admitted to regular ward / early discharged / admitted to subintensive units). However, this could be a potential implementation for future studies. We have added a sentence in the text in the study limitations section.

Round 2

Reviewer 2 Report

No further comments